# Purine nucleotide depletion prompts cell migration by stimulating the serine synthesis pathway

Mona Hoseini Soflaee[1,8], Rushendhiran Kesavan [1,8], Umakant Sahu[2,3,8], Alpaslan Tasdogan [4], Elodie Villa[2,3], Zied Djabari[2,3], Feng Cai[1], Diem H. Tran [1], Hieu S. Vu[1], Eunus S. Ali [2,3], Halie Rion[1], Brendan P. O'Hara[2,3], Sherwin Kelekar[1], James Hughes Hallett[5], Misty Martin[1], Thomas P. Mathews[1], Peng Gao [6], John M. Asara[7], Brendan D. Manning [5✉], Issam Ben-Sahra [2,3✉] & Gerta Hoxhaj [1✉]

Purine nucleotides are necessary for various biological processes related to cell proliferation. Despite their importance in DNA and RNA synthesis, cellular signaling, and energy-dependent reactions, the impact of changes in cellular purine levels on cell physiology remains poorly understood. Here, we find that purine depletion stimulates cell migration, despite effective reduction in cell proliferation. Blocking purine synthesis triggers a shunt of glycolytic carbon into the serine synthesis pathway, which is required for the induction of cell migration upon purine depletion. The stimulation of cell migration upon a reduction in intracellular purines required one-carbon metabolism downstream of de novo serine synthesis. Decreased purine abundance and the subsequent increase in serine synthesis triggers an epithelial-mesenchymal transition (EMT) and, in cancer models, promotes metastatic colonization. Thus, reducing the available pool of intracellular purines re-routes metabolic flux from glycolysis into de novo serine synthesis, a metabolic change that stimulates a program of cell migration.

[1] Children's Medical Center Research Institute, University of Texas Southwestern Medical Center, 5323 Harry Hines Boulevard, Dallas, TX 75390, USA.
[2] Department of Biochemistry and Molecular Genetics, Feinberg School of Medicine, Northwestern University, Chicago, IL 60611, USA. [3] Robert H. Lurie Comprehensive Cancer Center, Northwestern University, Chicago, IL 60611, USA. [4] Department of Dermatology, University Hospital Essen & German Cancer Consortium, Partner Site, Essen, Germany. [5] Department of Molecular Metabolism, Harvard T. H. Chan School of Public Health, Boston, MA 02115, USA. [6] Metabolomics Core Facility, Robert H. Lurie Comprehensive Cancer Center, Northwestern University, Chicago, IL 60611, USA. [7] Mass Spectrometry Core, Beth Israel Deaconess Medical Center, Department of Medicine, Harvard Medical School, Boston, MA 02115, USA. [8] These authors contributed equally: Mona Hoseini Soflaee, Rushendhiran Kesavan, Umakant Sahu. ✉email: bmanning@hsph.harvard.edu; issam.ben-sahra@northwestern.edu; gerta.hoxhaj@utsouthwestern.edu

**M**etabolic reprogramming in proliferating cells favors an anabolic state, which serves to fulfill the increased macromolecular demands of cell division[1]. Cancer cells, for instance, modulate flux through glycolysis at multiple control points to favor the accumulation of metabolites that feed into biosynthetic pathways, such as the pentose phosphate pathway, serine/glycine synthesis, and de novo nucleotide synthesis[2–5].

Nucleotides are essential for numerous core biological processes, including DNA replication, RNA synthesis, maintenance of cellular energy, and cellular signaling[6–9]. The mechanistic Target of Rapamycin Complex 1 (mTORC1) has emerged as a key regulator of pyrimidine and purine nucleotide synthesis to support anabolic cell growth[7–9]. Reciprocally, depletion of cellular purine nucleotides via antimetabolites that target nucleotide synthesis,

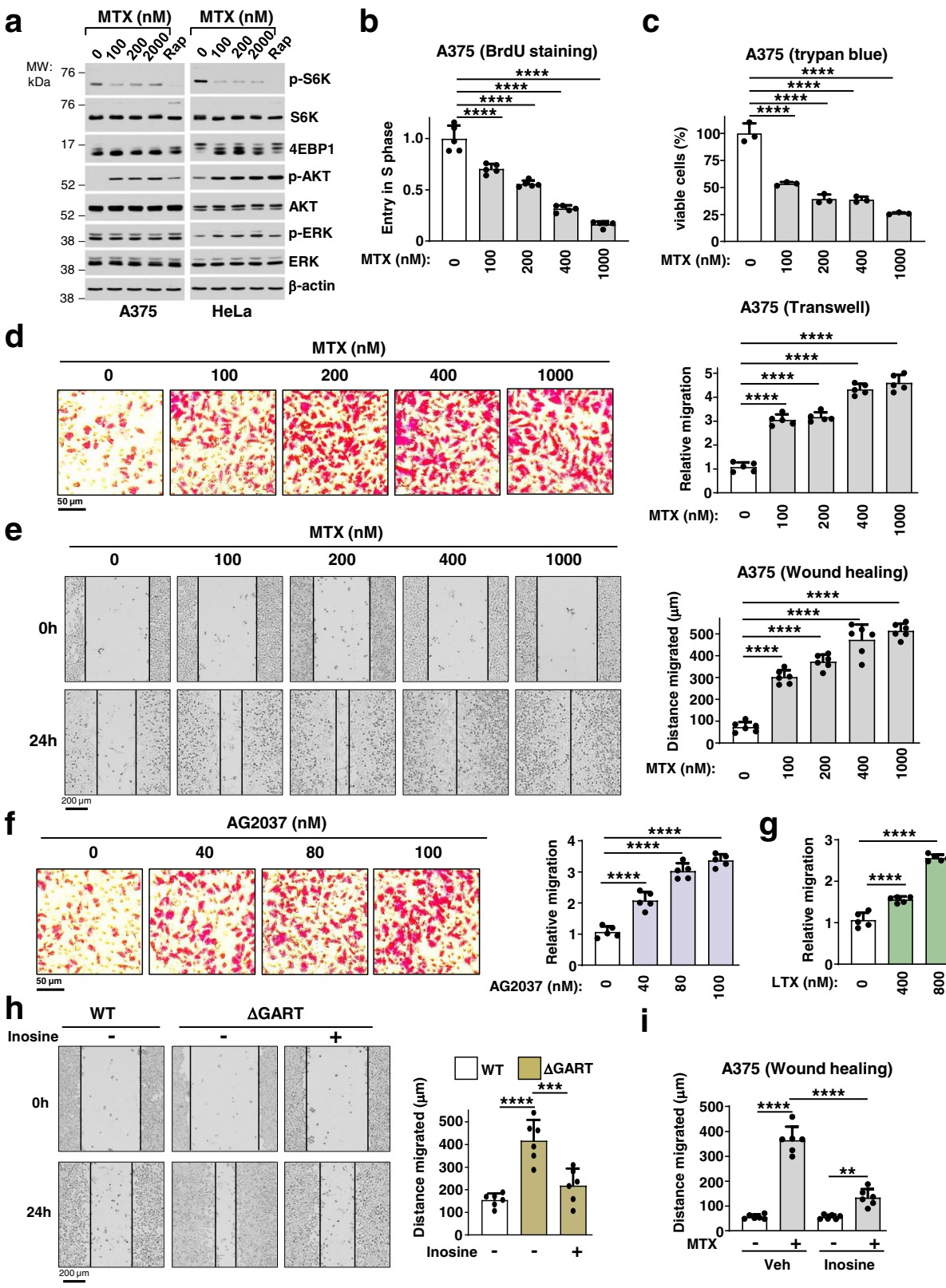

**Fig. 1 Purine depletion triggers cell migration despite a decrease in cell viability. a** Immunoblots of A375 and HeLa cells treated for 24 h with the indicated doses of MTX. **b, c** Cell-cycle progression through S phase (cell proliferation, BrdU incorporation) (**b**) and relative cell viability (Trypan blue) (**c**) of A375 cells treated with either vehicle (DMSO) or the indicated doses of MTX for 24 h. **d** Representative images of A375 cells after migration through a transwell (Boyden chamber) treated with either vehicle or the indicated doses of MTX. The migrated cells from five independent biological replicates are presented as relative migration normalized to control (untreated). **e** Representative images of A375 cells after a wound-healing assay, which were treated with the indicated doses of MTX for 24 h in A375 cells. Quantification of the migrated distance [Gap distance at time 0 h minus gap distance at 24 h] from 6 independent biological replicates. **f** Representative images of A375 cells after migration through a transwell treated with either vehicle or the indicated doses of GART inhibitor AG2037. The migrated cells were counted from five independent biological replicates and presented as relative migration normalized to control (untreated). **g** Quantification of the relative migration from a transwell assay in A375 cells treated with vehicle (DMSO) or with the indicated doses of LTX for 24 h as in (**f**). **h** Representative images from a wound-healing assay of wild-type and $\Delta$GART HeLa cells grown with or without inosine (50 μM) for 24 h. Quantification of the migrated distance was done as in **e**. **i** Quantification of the migrated distance from a wound-healing assay (24 h) as in **e**, in A375 cells treated for 24 h with MTX in the presence or absence of inosine (50 μM). **b–i** Data are presented as the mean ± s.d from $n = 3$ (**c**), or $n = 5$ (**b, d, f, g**) or $n = 6$ (**e, h, i**) of biologically independent samples. **b–i** One-way ANOVA, Turkey's post-hoc test, multiple comparison, **$p < 0.01$, ***$p < 0.001$, ****$p < 0.0001$. **d–f, h** Scale bars are indicated. **a–i** Data are representative of $n = 2$ (**a**), $n = 3$ (**b–i**) independent experiments. Source data and exact p-values are provided as a Source Data file.

such as methotrexate (MTX) or lometrexol (LTX), robustly inhibits mTORC1 activity[10,11], suggesting that cellular purines impact mTORC1-dependent cellular and metabolic functions. However, despite the variety of essential roles of purine nucleotides in processes related to cell proliferation and growth[6], our understanding of how cells sense and respond to changes in intracellular purine nucleotide concentrations remains limited.

Here, we systematically assess the effects of purine nucleotide levels on cell fitness and metabolism. Surprisingly, we found that chemical or genetic induction of purine depletion resulted in increased cell migration, despite a sturdy reduced proliferation rate. Through steady-state metabolite profiling and isotope tracing methods, we demonstrated that purine synthesis inhibition led to an increase in metabolic flux through de novo serine synthesis. This stimulated increase in de novo synthesized serine was essential for the promotion of cell motility and metastasis upon purine depletion. Thus, these findings demonstrate that a metabolic link between purine and serine metabolism has profound effects on cell migration and provides insights into the effects of pharmacological manipulation of these metabolic pathways to target the growth and metastatic spread of tumor cells.

## Results

**Inhibition of de novo purine synthesis triggers cell migration.** Since a decrease in cellular purine levels inhibits mTORC1 signaling[10,11], without inhibitory effects on ERK or Akt signaling (Fig. 1a), we sought to understand the impact of purine depletion on cellular physiology. To this end, we measured effects on cell proliferation, viability, and migration in response to increasing doses of the purine synthesis inhibitor MTX (Fig. 1a-e, Supplementary Fig. 1a,c). Treatment of A375 (melanoma) or HeLa (cervical cancer) cells with MTX, delayed cell-cycle entry into the S phase (Fig. 1b), and impaired cell viability (Fig. 1c; Supplementary Fig. 1a), albeit with no apparent induction of apoptotic or autophagic markers (Supplementary Fig. 1b, c). Surprisingly, MTX treatment concomitantly increased the migration of A375 cells in a dose-dependent manner, as measured by two distinct cell migration assays: transwell Boyden chamber and wound-healing assays (Fig. 1d, e). MTX-stimulated cell migration was also observed in a variety of other cell lines tested, including HeLa, SK-MEL-28, B16-F10, and CAL-51 (Supplementary Fig. 1d-i), and these effects were independent of the p53 status (Supplementary Fig. 1j, k). Importantly, other chemical inhibitors of purine synthesis, including AG2037[11] and LTX, structurally distinct compounds that target the purine synthesis enzyme phosphoribosylglycinamide formyltransferase (GART), also stimulated cell migration (Fig. 1f, g). The effects of purine depletion on cell migration were independent of its

inhibitory effects on mTORC1, as the mTORC1 inhibitor rapamycin decreased cell migration (Supplementary Fig. 2a). Genetic perturbation of purine synthesis via CRISPR/Cas9-mediated knockout of GART ($\Delta$GART) also increased HeLa cell migration (Fig. 1h, Supplementary Fig. 2b). Exogenous addition of the purine nucleoside inosine significantly attenuated the enhanced migration of $\Delta$GART HeLa cells and MTX-treated A375 cells (Fig. 1h, i; Supplementary Fig. 2c,d), indicating that cell migration is induced as a result of decreased purine availability.

**Purine depletion raises serine synthesis intermediates.** To gain insight into the mechanisms by which purine depletion triggers increased cell migration, we hypothesized that purine insufficiency reprograms metabolism to stimulate cell motility. We performed unbiased steady-state metabolite profiling in HeLa cells treated with various nucleotide synthesis inhibitors targeting either purine or pyrimidine synthesis (Fig. 2a). Interestingly, inhibition of purine synthesis with MTX or LTX resulted in increased intracellular levels of serine and 3-phosphoserine (3-PS), a specific intermediate in the de novo serine synthesis pathway (Fig. 2a), which was not observed upon inhibition of the de novo pyrimidine synthesis enzymes dihydroorotate dehydrogenase (DHODH), with leflunomide, or thymidylate synthase (TYMS), with 5-FU, or mTORC1 with rapamycin (Fig. 2a; Supplementary Fig. 3a). Similarly, siRNA-mediated knockdown of the purine synthesis enzyme GART, but not the pyrimidine synthesis enzyme DHODH, also resulted in an increase in the 3-PS and serine levels (Fig. 2b; Supplementary Fig. 3b). A time course of MTX treatment revealed that concomitant with a global decrease in purine nucleotides, 3-phosphoserine levels are increased within 4 h of treatment and continued to increase with time (Fig. 2c, d). This effect was seen in all six cell lines tested, with MTX treatment increasing 3-PS levels in LNCaP, SK-MEL-28, A375, and A549 cells, indicating a generalizable effect of purine depletion in altering the serine synthesis pathway (Fig. 2e). Mice subjected to MTX treatment also showed an increase in both 3-PS and serine in the spleen and liver, suggesting that increased 3-PS levels are a metabolic biomarker of purine depletion in cultured mammalian cells and in vivo (Fig. 2f).

To determine whether a decrease in purine levels is directly linked to the increase in 3-PS and serine levels observed, we replenished purine pools with an exogenous nucleoside (inosine) or nucleobase (adenine) following MTX or LTX treatment or GART deletion (Fig. 2g; Supplementary Fig. 3c). Supplementation with inosine for just one hour was able to restore both IMP and 3-PS levels under these conditions (Fig. 2h; Supplementary Fig. 3c). To determine whether exogenous inosine needs to be converted into nucleotides to rescue 3-PS and serine levels, we used a purine

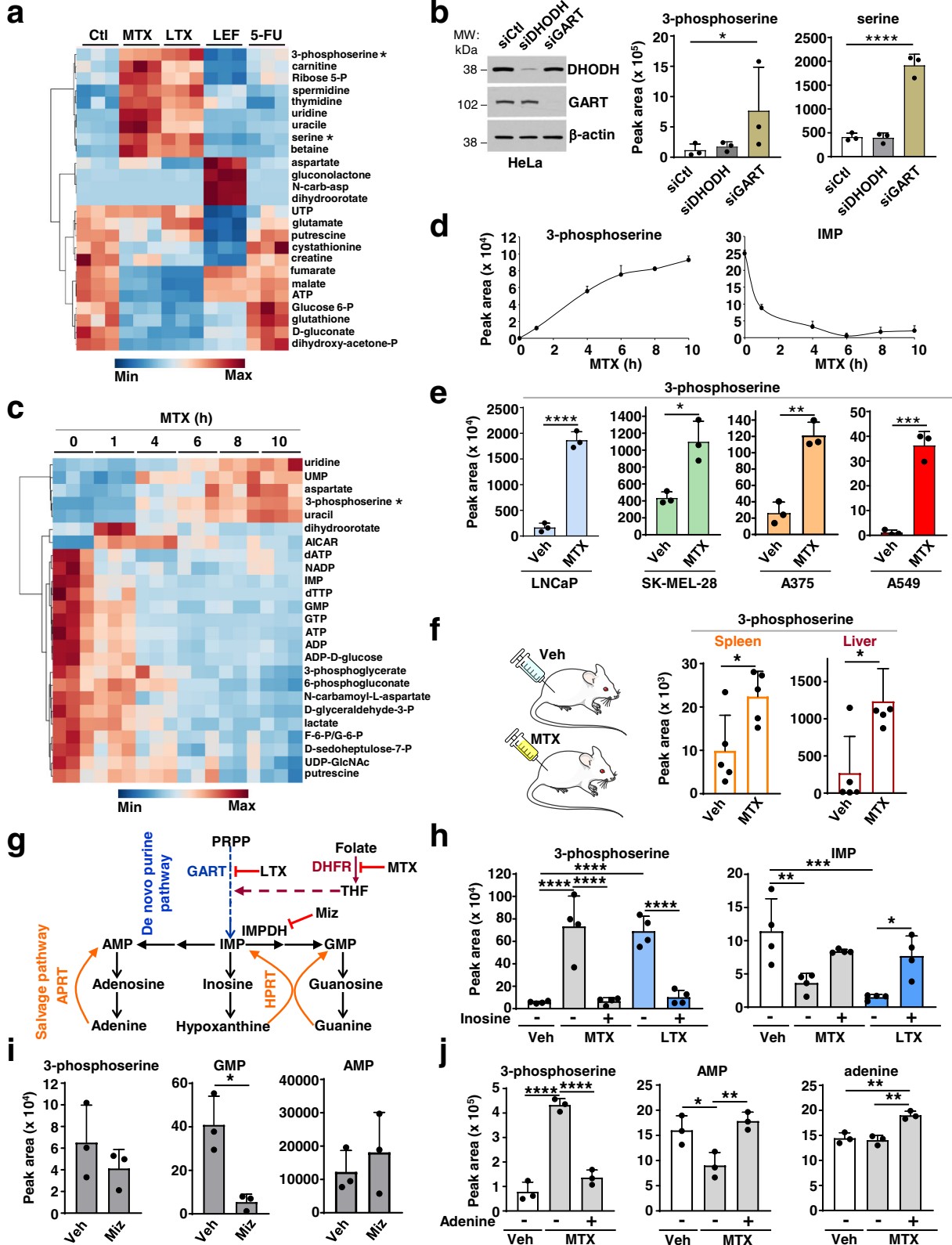

salvage-deficient fibroblast cell line that lacks both of the purine salvage pathway enzymes, hypoxanthine phosphoribosyltransferase (HPRT) and adenine phosphoribosyltransferase (APRT), and thus cannot convert purine nucleosides or nucleobases into nucleotides[10]. We found that while MTX treatment significantly increased 3-PS levels in both $Hprt^{-/-};Aprt^{-/-}$ cells (A9) and their wild-type counterparts (CCL1), exogenous inosine restored both

IMP and 3-PS only in wild-type cells, indicating that inosine conversion to purine nucleotides is essential for its acute suppressive effects on the MTX-mediated increase in 3-PS (Supplementary Fig. 3d). To test whether the increase in 3-PS and serine levels is mediated by depletion of adenylates or guanylates, we specifically targeted guanylate production by inhibiting inosine monophosphate dehydrogenase (IMPDH), the rate-limiting step in de novo

**Fig. 2 Depletion of purines results in elevated 3-phosphoserine and serine levels. a** Steady-state metabolite profile from HeLa cells treated for 15 h with DMSO (Ctl), MTX (2 μM), LTX (2 μM), Lef (10 μM), and 5-FU (1 μM). **b** Normalized peak areas of 3-phosphoserine and serine measured by LC-MS/MS from HeLa cells transfected with nontargeting controls (siCtl), or siRNA against DHODH or GART. **c** As in (**a**) from HeLa cells treated with MTX (2 μM) for the indicated times. **d** Normalized peak areas of 3-phosphoserine and IMP from HeLa cells treated with MTX (2 μM) for the indicated times are shown from the experiment in (**c**). **e** Normalized peak areas of 3-phosphoserine after vehicle (Veh, DMSO) or MTX treatment (2 μM, 15 h) from LNCaP (prostate cancer), SK-MEL-28 (melanoma), A375 (melanoma), and A549 (lung cancer) cells. **f** Normalized peak areas of 3-phosphoserine from murine spleen and liver after daily treatment with vehicle (0.9% saline) or MTX (40 mg/kg) for five consecutive days. **g** Schematic of the de novo and salvage purine synthesis pathways. The targets of the purine synthesis inhibitors (MTX or LTX) and Mizoribine (Miz) are indicated. **h** Normalized peak areas of 3-phosphoserine and IMP after treatment of HeLa cells treated with vehicle (Veh, DMSO), MTX (2 μM), or LTX (2 μM) for 15 h, followed by treatment with inosine (50 μM, 1 h). **i** Normalized peak areas of the indicated metabolites from HeLa cells treated with vehicle (Veh, DMSO) or Mizoribine (Miz, 15 μM) for 15 h. **j** Normalized peak areas of the indicated metabolites from HeLa cells treated with vehicle (Veh, DMSO) or MTX (2 μM) for 15 h, followed by treatment with adenine (50 μM, 1 h). **b**, **d**–**f**, **h**–**j** Data are presented as the mean ± s.d from n = 3 (**b**, **d**, **e**, **i**, **j**) or n = 4 (**h**), or n = 5 (**f**) of biologically independent samples. Data are presented from n = 3 independent samples (**a**, **c**). **b**, **d**, **h**, **j** One-way ANOVA, Turkey's post-hoc test, multiple comparison, **e**, **f**, **i** Unpaired t-test, *p < 0.05, **p < 0.01, ***p < 0.001, ****p < 0.0001. Data are representative of n = 2 (**a**–**d**, **f**, **i**, **j**), n = 3 (**e**), n = 4 (**h**) independent experiments. Source data and exact p-values are provided as a Source Data file.

guanylate synthesis[12]. Depletion of guanylates with the IMPDH inhibitor mizoribine, without effects on adenylates, did not increase 3-PS levels (Fig. 2i). Conversely, the addition of adenine alone, which is metabolized into AMP by APRT, completely restored 3-PS levels after MTX treatment (Fig. 2j), indicating that the increase in 3-PS and serine observed is the specific result of adenylate depletion.

**Adenylate depletion stimulates flux into serine synthesis**. To determine whether the robust increase in 3-PS and serine levels upon purine depletion reflected increased metabolic flux through the de novo serine synthesis pathway, we employed stable isotope tracing with $^{13}C_6$-glucose (Fig. 3a). Similar to the steady-state measurements, MTX treatment stimulated an increase in the synthesis of 3-PS (M + 3) and serine (M + 3) from glucose, an effect that was completely suppressed by adenine supplementation (Fig. 3b). Similarly, adenine addition also rescued the increased levels of 3-PS (M + 3) and serine (M + 3) in GART deleted cells (Supplementary Fig. 4a). To confirm the specificity and directionality of the metabolic tracing into newly synthesized 3-PS and serine, we treated cells with two structurally distinct inhibitors of phosphoglycerate dehydrogenase (PHGDH), the rate-limiting enzyme in the serine synthesis pathway (Fig. 3a, c). PHGDH inhibition blocked the glucose-derived synthesis of 3-PS and serine in both vehicle- and purine-depleted-treated cells, indicating that PHGDH activity required the purine depletion-dependent increase in serine synthesis (Fig. 3c,d; Supplementary Fig. 4b, c). Moreover, inhibition of the synthesis of the PHGDH substrate 3-phosphoglycerate with koningic acid, a selective inhibitor of glyceraldehyde 3-phosphate dehydrogenase (GAPDH)[13], also fully blocked de novo serine synthesis both at the basal levels and in response to purine depletion (Fig. 3e). Despite the stimulated increase in serine synthesis, the protein abundance of the serine synthesis enzymes was not affected by MTX or LTX treatment (Supplementary Fig. 4d), and the catalytic activity of PHGDH was not altered by the direct addition of various nucleotide species to the enzymatic assay (Supplementary Fig. 4e,f).

The increase in serine synthesis upon purine depletion was not due to a general increase in glycolytic flux (Supplementary Fig. 4g,h). The MTX- and LTX-induced increase in flux into the serine synthesis pathway was accompanied by a significant decrease in the extracellular acidification rate (ECAR) and glucose-derived lactate, without substantial changes in oxygen consumption rate (OCR) (Fig. 3f-h; Supplementary Fig. 4i). Furthermore, GART deletion also decreased the glycolytic conversion of glucose into lactate (Fig. 3g; Supplementary Fig. 4j).

This metabolic phenotype—decreased glycolytic flux accompanied by an increased flux into the de novo serine synthesis pathway—is reminiscent of effects associated with inhibition of the final enzyme of glycolysis, pyruvate kinase M (PKM)[14], with PKM2 isoform being the dominant species in proliferating cells that possess unique biochemical properties tailored for this role[15]. Moreover, MTX treatment also resulted in the decrease of the pyruvate to phophoenolpyruvate (PEP) ratio, which was restored by exogenous adenine (Supplementary Fig. 4k), suggesting a purine-dependent regulation of the pyruvate kinase step. Depletion of purines with MTX led to a greater than two-fold decrease in the intracellular concentrations of ATP (from ~14 to ~5 mM) and ADP (~120 to ~50 μM) (Supplementary Fig. 4l). Given that the $K_d$ of PKM2 for ADP is 200–300 μM[16], its activity would be sensitive to this degree of ADP depletion. Indeed, we found that loss of *Pkm2* (Δ*Pkm2*) in MEFs exhibit an increased shunt from glycolysis into serine synthesis relative to wild-type MEFs, and MTX treatment failed to increase serine flux in these cells despite a robust depletion of purine nucleotides Fig. 3i, j; Supplementary Fig. 4m). Reciprocally, the selective PKM2 activator TEPP-46, which induces the formation of the active PKM2 tetramer[17,18], blocked the ability of MTX to stimulate serine flux (Fig. 3k). Thus, PKM2 is both necessary and sufficient to control flux through de novo serine synthesis in response to purine depletion.

**Serine and one-carbon metabolism promote cell migration**. We next asked whether the increased flux into the serine synthesis pathway upon purine depletion influences the coincident effects on cell migration. Three structurally distinct PHGDH inhibitors reduced A375 cell viability by 30–40%, but when combined with MTX did not further reduce viability relative to MTX treatment alone (Supplementary Fig. 5a). The PKM2 activator TEPP-46 also did not show any significant additive effects on cell viability when combined with MTX (Supplementary Fig. 5b). However, the MTX-mediated increase in cell migration was abolished by blocking the stimulated flux into serine synthesis through either PHGDH inhibition (Fig. 4a; Supplementary Fig. 5c) or PKM2 activation (Fig. 4b; Supplementary Fig. 5d). Next, we employed a genetic approach to assess the effects of serine synthesis on cell migration by generating A375 and HeLa cell lines with CRISPR/Cas9-mediated *PSAT1* or *PHGDH* knockouts (Δ*PSAT1* and Δ*PHGDH*) (Supplementary Fig. 5e, f). As anticipated, Δ*PHGDH* cells exhibit a strong decrease in 3-PS and serine levels and were auxotrophic for serine (Supplementary Fig. 5g, h). Interestingly, MTX treatment failed to promote cell migration in Δ*PSAT1* HeLa cells (Fig. 4c) and in Δ*PHGDH* A375 cells (Fig. 4d, e), suggesting that endogenously produced serine is required to boost cell migration in response to purine synthesis inhibition. Consistent

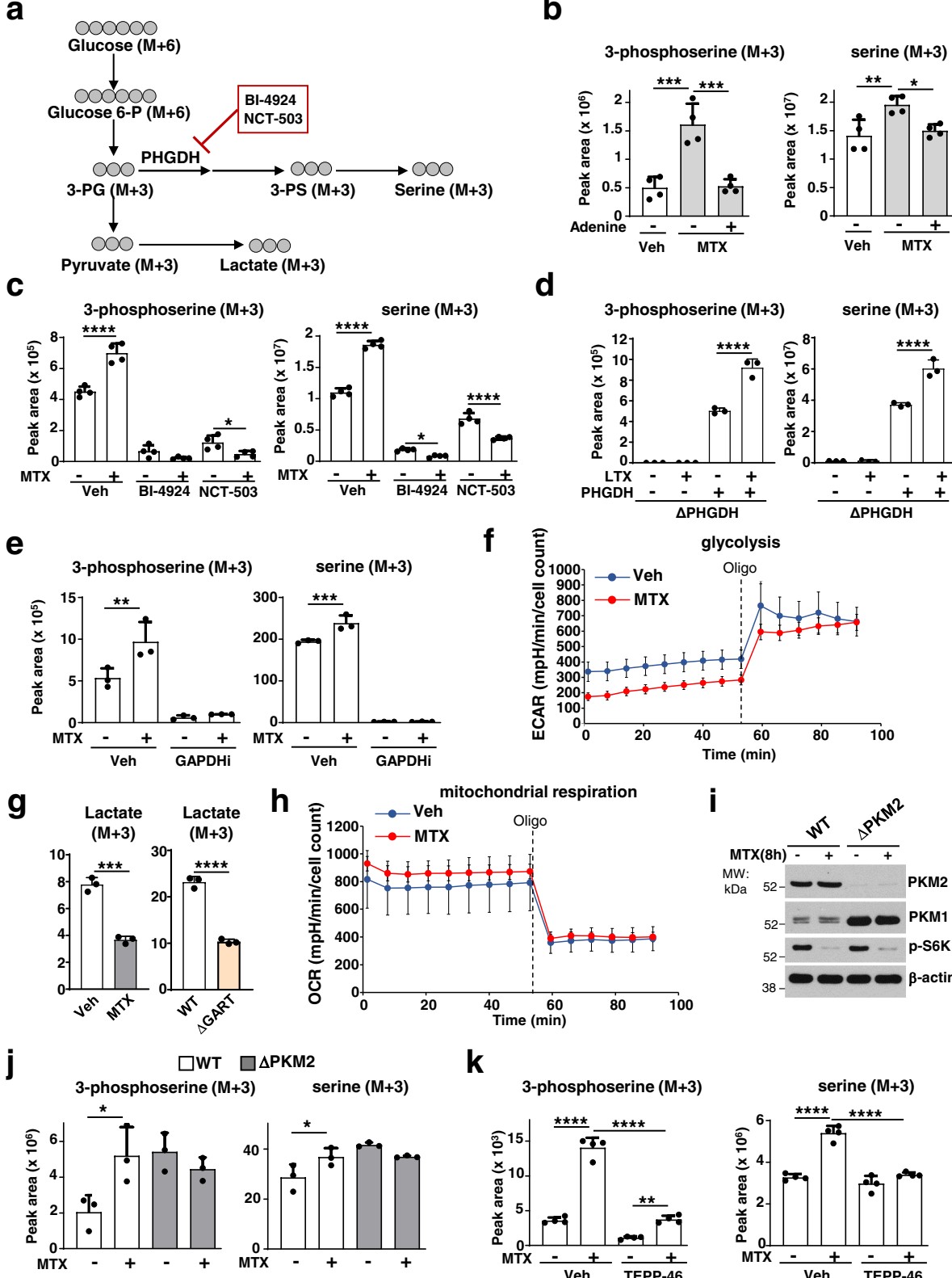

with this notion, we observed a partial restoration of MTX-induced migration in these knockout cells with higher concentrations of serine in the media (Fig. 4c,e) and a dose-dependent stimulatory effect of exogenous serine on the migration of untreated wild-type cells (Fig. 4f; Supplementary Fig. 5i).

While purine depletion was previously unknown to stimulate cell migration in a manner dependent on increased flux through the serine synthesis pathway, serine-derived formate produced through one-carbon metabolism (Fig. 4g) has been reported to promote cell invasion[19]. Akin to serine supplementation, formate addition also resulted in increased cell migration in A375 and HeLa cells (Fig. 4h; Supplementary Fig. 5j). Inhibition of serine entry into one-carbon derivatives via the SHMT1/2 inhibitor SHIN1[20], or direct inhibition of formate synthesis through

**Fig. 3 Adenylate depletion promotes metabolic flux through the de novo serine synthesis pathway. a** Schematic of carbon flow from $^{13}C_6$-glucose into glycolytic and serine synthesis intermediates. PHGDH inhibitors: BI-4924, NCT-503. **b** Normalized peak areas of 3-phosphoserine (M + 3) and serine (M + 3) from HeLa cells treated with either veh (DMSO) or MTX (2 μM, 15 h), followed by adenine supplementation (50 μM, 1 h), and labelled with $^{13}C_6$-glucose in the last hour. **c** As in **b**, but from HeLa cells treated with MTX (2 μM, 15 h), in the presence or absence of BI-4924 (15 μM), or NCT503 (10 μM), and labelled for the last 3 h with $^{13}C_6$-glucose. **d** As in **b**, but from ΔPHGDH HEK293E cells reconstituted or not with PHGDH cDNA, treated with LTX (4 μM, 8 h), and labelled with $^{13}C_6$-glucose (1 h). **e** As in **b**, but from HeLa cells treated with MTX (2 μM) in the presence or absence of GAPDH inhibitor heptelidic acid (75 μM, 15 h), and labelled with $^{13}C_6$-glucose (3 h). **f** Extracellular acidification rate (ECAR) of HeLa cells treated with veh (DMSO) or MTX (2 μM, 15 h). **g** Normalized peak areas of lactate from HeLa cells treated with MTX (2 μM) (Left panel) or from wild-type or (ΔGART) HeLa cells grown without nucleotides for 24 h. **h** Oxygen consumption rate (OCR) of HeLa cells treated with MTX (2 μM, 15 h). **i** Immunoblots of wild-type and knockout PKM2 (ΔPKM2) MEFs, treated with veh (DMSO) or MTX (2 μM, 8 h). **j** As in **b**, but from wild-type and ΔPKM2 MEFs treated as in **i** and labelled with $^{13}C_6$-glucose (3 h). **k** As in **b**, but from HeLa cells that were pretreated with TEPP-46 (100 μM, 1 h) prior to treatment with MTX (2 μM, 8 h), and labelled with $^{13}C_6$-glucose (3 h). **b–h**, **j**, **k** Data are presented as the mean ± s.d from $n = 3$ (**d**, **e**, **g**, **j**), $n = 4$ (**b**, **c**, **k**), or $n = 6$ (**f**, **h**) of biologically independent samples. **b–e**, **j**, **k** One-way ANOVA, Turkey's post-hoc test, multiple comparison, **g** Unpaired $t$-test, *$p < 0.05$, **$p < 0.01$, ***$p < 0.001$, ****$p < 0.0001$. Data are representative of $n = 2$ (**d**, **f**, **g**, **h**) or $n = 3$ (**b**, **c**, **e**, **i-k**) independent experiments. Source data and exact $p$-values are provided as a Source Data file.

MTHFD1L knockdown, significantly reduced the MTX-mediated increase in cell migration (Fig. 4i, j; Supplementary Fig. 5k-m), indicating that the downstream conversion of serine to one-carbon units and formate contribute to the enhanced cell migration induced by purine depletion (Fig. 4k).

**Purine depletion and serine synthesis trigger an EMT program.** To gain insight into the molecular mechanisms that control cell migration downstream of changes in purine and serine synthesis, we first examined the organization of the actin cytoskeleton, which provides internal mechanical support for cell shape and motility[21]. Phalloidin staining of F-actin filaments showed marked elongation of cells in response to MTX treatment (Fig. 5a). These changes were purine-dependent, as inosine addition restored normal cell morphology (Fig. 5a). We obtained similar results with two other selective inhibitors of de novo purine synthesis, LTX and AG2037, indicating a link between purine abundance and cell morphology (Fig. 5b). These morphological changes led us to assess transcriptional changes associated with the epithelial-mesenchymal transition (EMT), which often precedes enhanced migratory properties of cells[22]. Interestingly, we observed that treatment with various purine synthesis inhibitors, including MTX, LTX, and AG2037, triggers an EMT transcriptional program characterized by N-cadherin and Vimentin upregulation and E-cadherin downregulation, indicative of a shift toward a mesenchymal cell fate[23,24] (Fig. 5c; Supplementary Fig. 6a,b). Like cell elongation and migration, this change in EMT markers upon depletion of purines was blocked with supplementation of exogenous inosine (Fig. 5c).

Given its essential nature for the stimulated cell migration, we next determined whether the increase in serine synthesis triggered by purine depletion was required for these effects on cell morphology and EMT. Indeed, reducing serine synthesis with a PHGDH inhibitor or PKM2 activator prevented the MTX-induced effects on the actin cytoskeleton, cell length, and EMT (Fig. 5d-g; Supplementary Fig. 6c-e). Furthermore, MTX treatment was unable to induce an EMT phenotype in ΔPHGDH A375 cells (Fig. 5f; Supplementary Fig. 6d). Finally, serine or formate supplementation, alone, was sufficient to increase cell length and stimulate the expression of EMT markers (Fig. 5h, i). These data indicate that purine and serine abundance influence pro-migratory effects on cell morphology and the EMT transcriptional program.

**Methotrexate induces PKM2-dependent metastatic colonization.** EMT and enhanced migration are hallmarks of cancer cell metastasis and spread to distant sites[22,25,26]. To examine whether MTX can induce metastasis, we measured melanoma cell dissemination in mice from a primary tumor to secondary organ sites, such as lymph nodes and lungs. We treated mice bearing subcutaneous tumors derived from B16-F10 murine melanoma cells with vehicle or MTX for 3 weeks (Fig. 6a). While MTX treatment did not significantly affect primary tumor growth (Fig. 6b), it increased the spread of melanoma cells to the lymph nodes and lungs (Fig. 6c-f). Furthermore, MTX treatment increased serine and 3-phospho-serine levels in the blood, confirming its effect on serine metabolism in vivo (Fig. 6g). These data suggest that MTX promotes the metastatic potential of melanoma cells, an effect accompanied by an increase in circulating serine levels.

To determine whether MTX treatment influences distant colonization of melanoma cells and the role of increased serine synthesis driven by PKM2 inhibition, A375 cells were injected intravenously, and mice were treated with vehicle or MTX in the presence or absence of the PKM2 activator TEPP-46[17] (Fig. 6h). Notably, mice treated with MTX showed an increase in the number and size of micrometastases in their lungs (Fig. 6i-k), an effect blocked by co-treatment with the PKM2 activator. Importantly, metabolite profiling of the lungs of these mice showed a corresponding increase in serine and 3-phospho-serine levels upon MTX treatment that was also prevented by TEPP-46 treatment (Fig. 6l). These findings indicate that MTX treatment enhances the colonization of melanoma cells in a manner dependent on PKM2 inhibition and associated with changes in serine metabolism.

## Discussion

Our findings indicate that the de novo serine synthesis pathway and cell migration are induced as part of a metabolic and cellular adaptation to purine insufficiency (Fig. 4k). Purine depletion mirrored the metabolic effects canonically observed in response to PKM2 inactivation, causing a general decrease in glycolytic flux while concomitantly increasing the shunt of glycolytic intermediates through the de novo serine synthesis pathway[18]. Our data demonstrate that PKM2 is required for the induction of serine synthesis upon purine depletion. A functional connection between serine and PKM2 has been established previously, where serine starvation was found to inhibit PKM2 activity[27], resulting in increased metabolic shunt into de novo serine synthesis, thus enabling cancer cell survival under limited serine conditions. Our results support a model in which purine scarcity inhibits PKM2 activity, likely by limiting the availability of its substrate ADP, leading to glucose-derived carbon being channeled from glycolysis into serine biosynthesis, thereby reprogramming metabolism and altering cellular behavior to enable cells to adapt to purine nucleotide stress[28].

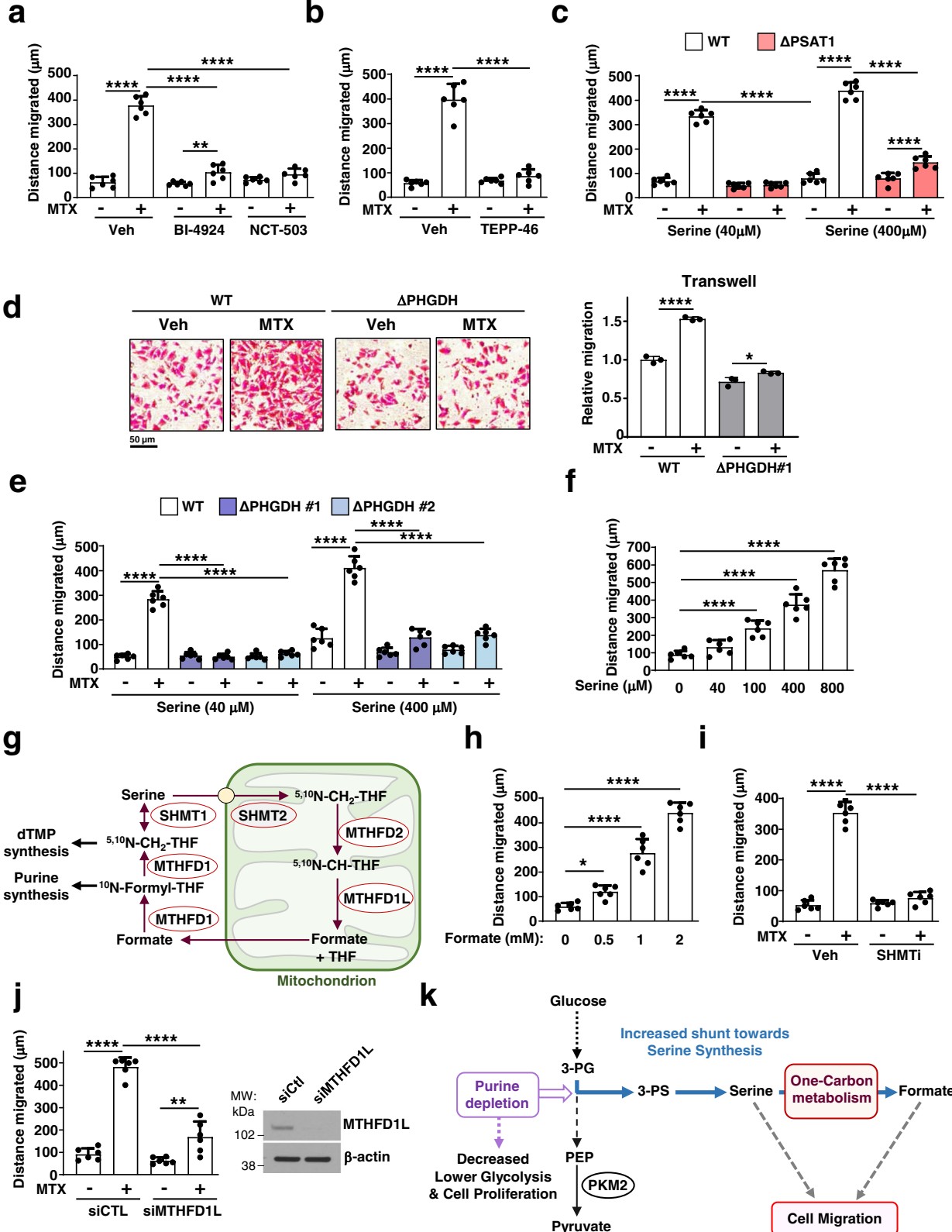

This study also revealed that de novo serine synthesis and one-carbon metabolism promote cell migration in response to purine depletion. Interestingly, supplementation with serine or formate (generally produced by one-carbon metabolism) was sufficient to induce cell migration, indicating a critical role for serine-derived formate in mediating cancer cell migration[19]. Furthermore, the pro-migratory phenotype associated with purine depletion or augmented serine/one-carbon metabolism was accompanied by changes in actin dynamics, increased cell length, and stimulation of an EMT program, which promotes the invasive ability and dissemination of cancer cells.

Purine and pyrimidine synthesis inhibitors such as MTX and 5-FU have been used routinely as chemotherapeutic agents in various cancers for the past 60 years, with somewhat successful

**Fig. 4 Purine depletion-induced cell migration is dependent on serine synthesis and one-carbon metabolism. a** Quantification of the migrated distance (Wound-healing assay, 24 h) of A375 cells after MTX treatment (100 nM) in the presence or absence of PHGDH inhibitors (BI-4924, 15 μM and NCT-503, 10 μM). **b** As in (**a**), but from A375 cells after MTX treatment (100 nM) in the presence or absence of TEPP-46 (100 μM). **c** As in **a**, but from wild-type or PSAT1 knockout (ΔPSAT1) HeLa cells after MTX treatment (100 nM) in growth media with 40 μM or 400 μM of serine. **d** Representative images and quantification of wild-type or ΔPHGDH A375 cells (transwell assay) treated with either veh or MTX (200 nM) cultured in 40 μM of serine. Relative migration normalized to control (untreated). **e** Quantification of the migrated distance (Wound-healing assay, 24 h) of wild-type or PHGDH knockout (ΔPHGDH, clones #1, #2) A375 cells after MTX treatment (100 nM), cultured in 40 μM or 400 μM serine. **f** As in (**a**), but from A375 cells after treatment with the indicated concentrations of serine in serine/glycine-free growth media. **g** Schematic showing serine catabolism and the one-carbon folate cycle. dTMP deoxythymidinemonophosphate, THF tetrahydrofolate. **h** As in (**a**), but from A375 cells after treatment with the indicated doses of formate. **i** As in (**a**), but from A375 cells treated with veh or MTX (100 nM) in the presence or absence of SHMT1/2 inhibitor (SHIN1, 5 μM). **j** As in (**a**), but from A375 cells transfected with nontargeting controls (siCtl) or siRNA targeting MTHFD1L and treated with veh or MTX (100 nM). **k** Model of purine depletion stimulating de novo serine synthesis and cell migration, despite a reduced glycolytic flux and cell proliferation. **a–f**, **h–j** Data are the mean ± s.d from $n = 3$ (**d**) and $n = 6$ (**a–c**, **e**, **f**, **h–j**) of biologically independent samples. **a–f**, **h–j** One-way ANOVA, Turkey's post-hoc test, multiple comparison, *$p < 0.05$, **$p < 0.01$, ***$p < 0.001$, ****$p < 0.0001$. Data are representative of $n = 2$ (**d**, **j**) or $n = 3$ (**a–c**, **e**, **f**, **i**) independent experiments. Western blots are representative of $n = 2$ (**j**) independent experiments. **d** Scale bar is indicated. Source data and exact $p$-values are provided as a Source Data file.

clinical outcomes[29]. Our findings that specific inhibition of purine synthesis with MTX can trigger an EMT transcriptional program, cell migration phenotype in vitro, and enhanced metastatic dissemination in mouse models, suggest that MTX may induce cancer cell invasion. However, further work is required to elucidate whether MTX promotes specific cancer invasion features, enabling tumor cells to overcome barriers of the extracellular matrix and spread into surrounding tissues[30]. Similar to our findings, the support of cancer cell motility, migration, and metastasis needed the increase of de novo serine synthesis and mitochondrial one-carbon metabolism[31].

Consistent with these findings that induction of serine synthesis is essential for these effects of MTX, recent studies found that inhibition of the de novo serine synthesis enzyme PHGDH attenuates cancer metastasis[32,33]. Thus, our study further demonstrates that pharmacological inhibitors that alter serine levels can influence the capacity of cancer cells to spread. Our work also prompts further investigation on the potential of combining MTX with inhibitors of serine metabolism as a treatment strategy to hamper both cancer cell growth and invasive migration.

In summary, this study adds serine synthesis and cell migration as biological processes closely linked to intracellular purine availability. Our investigation provides a framework in which changes in cellular purine levels acutely alter the activity of central carbon metabolism to spur an adaptive morphological and migratory response.

## Methods

**Cell culture**. PKM2 knockout mouse embryonic fibroblasts MEFs and their matched wild-type counterparts were obtained from the laboratory of Dr. Matthew Vander Heiden (Massachusetts Institute of Technology, Cambridge, MA[34]. Standard cancer cell lines (A375, SK-MEL-28, LNCaP, CAL-51, A549, B16, and HeLa), Aprt$^{-/-}$ Hprt$^{-/-}$ A9 mouse fibroblasts (CCL-1.4) and their parental cell line (CCL-1) were from ATCC. All cell lines were maintained in DMEM (Corning/Cellgro, 10-017-CV) containing 10% fetal bovine serum (FBS) at 37 °C and 5% CO$_2$. Treatments with purine and pyrimidine inhibitors were performed in the presence of 10% dialyzed FBS (Sigma, F0392) to avoid the presence of nucleosides and nucleobases. Cells were treated with MTX, LTX, leflunomide, 5-fluorouracil (5-FU), 6-MP, mizoribine, TEPP-46, BI-4984, NCT-503, and SHIN1 as indicated in the figure legends. Exogenous adenine and inosine were added at 50 μM concentrations for 1 h unless otherwise indicated.

## Mouse studies

*Measurement of 3-phosphoserine in vivo*. All mice were fed a chow diet (Envigo, Cat # 2916) ad libitum and maintained in a pathogen-free environment with a 12:12 light/dark cycle. The temperature in the animal facility is kept at 72 F, with a range from 68 to 79 F, and humidity is kept at 33%, with a range between 30 and 70%. Six-week-old female BALB/cJ (Jackson Laboratory) (5 mice) were injected intraperitoneally with methotrexate (MTX, 40 mg/kg) or vehicle (0.9% saline) daily for 5 days (Fig. 2f). Mice were then sacrificed, and liver and spleen were collected and frozen immediately in liquid nitrogen. Metabolites were extracted in 80%

methanol, and details are provided in the metabolite profiling section. All animal procedures required for this study were accepted by the Institutional Animal Care and Use Committee (IACUC) at Northwestern University and were performed in accordance with relevant guidelines and regulations (Protocol no: IS00013905).

*Xenograft experiments*. All animal procedures were performed in accordance with ethical regulations and guidelines approved by the Institutional Animal Care and Use Committee (AICUC) at University of Texas Southwestern Medical Center (Protocol no: 2020–102880). A375 cells (50,000 cells) were injected into the tail vein of female athymic nude mice (8-week-old). Five days post injections, mice were treated either with vehicle (PBS) or low dose methotrexate (MTX; 20 mg/kg) three times a week via the intraperitoneal (IP) route. TEPP-46 (50 mg/kg) was provided daily through oral gavage. After ten weeks of treatment mice were euthanized. Lung tissue was collected for metabolomics and immunohistochemistry.

B16-F10 melanoma cells (100,000 cells) were injected subcutaneously in female athymic nude mice (8 weeks old). Once the tumors became palpable, mice were then treated with either vehicle (PBS), or MTX (20 mg/kg, IP, three times a week for three weeks. Tumor growth rates were monitored weekly with a caliper, and mice were euthanized once the first observed tumor reached a diameter of ~2 cm. Our approved animal protocol permits a maximum tumor size of 2.5 cm in diameter for melanoma metastasis studies. The maximum tumor size/burden was not exceeded. After euthanasia, axillary lymph nodes, lungs, and blood were collected for metabolite extraction and immunohistochemistry. The experimental design and setup were adopted from previous studies[35,36].

**RNAi, cDNA constructs, and CRISPR/Cas9**. All siRNAs were obtained from Dharmacon (ON-TARGETplus SMARTpool). siRNA-mediated knockdowns were performed according to the manufacturers' instructions. Cells plated in six-well plates were transfected with 20 nM ON-TARGETplus siRNA pools (Dharmacon) using Lipofectamine RNAiMAX (ThermoFisher Scientific, 13778). cDNAs of PHGDH (Cat #: MR224471), PSAT1 (Cat #: MR205716), and PSPH (Cat #: MR202626) expressing a C-terminal FLAG tag were obtained from Origene.

CRISPR/Cas9 knockout of human PHGDH, PSAT1, and GART sgRNA were generated in A375 or HeLa cells. sgRNA sequences for these genes were designed using CRISPOR (http://crispor.tefor.net) and cloned into a GFP-Cas9 expressing vector (PX458, Addgene, #48138). Cells were transfected with CRISPR/Cas9 sgRNA expressing plasmids, and the GFP-positive cells were single-cell sorted into 96-well plates via flow cytometry with BD FACS Aria Fusion. PHGDH and PSAT1 cells were grown in DMEM containing serine (400 μM), while GART cells were grown in DMEM supplemented with inosine (100 μM). Colonies were screened for the knockout by western blotting using respective antibodies. The following sequences were used:

sgPHGDH Sense: CACCGTGCAAGATCTTCCGGCAGCA,
sgPHGDH Antisense: AAACTGCTGCCGGAAGATCTTGCAC,
sgPSAT1 Sense: CACCGAAAGTTGACCACCTGCCTGG,
sgPSAT1 Antisense: AAACCCAGGCAGGTGGTCAACTTTC,
sgGART Sense: CACCGAAGCCTTCCATTGTGCGGTT,
sgGART Antisense: AAACAACCGCACAATGGAAGGCTTC
sgTP53 Sense (Exon 5): CACCG AGTGGAAGGAAATTTGCGTG
sgTP53 Antisense (Exon 5): AAACCACGCAAATTTCCTTCCACTC
sgTP53 Sense (Exon 6): CACCGACACATGTAGTTGTAGTGGA
sgTP53 Antisense (Exon 6): AAACTCCACTACAACTACATGTGTC

**mRNA expression analysis**. RNA extraction was performed using a RNeasy Plus Mini Kit (QIAGEN, 74136). RNA (1 μg) was subjected to reverse transcription with EcoDry Premix (Takara, 639545). The cDNA was diluted to 1:5 ratio with nuclease-free water and amplified using Bio-Rad S soAdvanced Universal SYBR Green

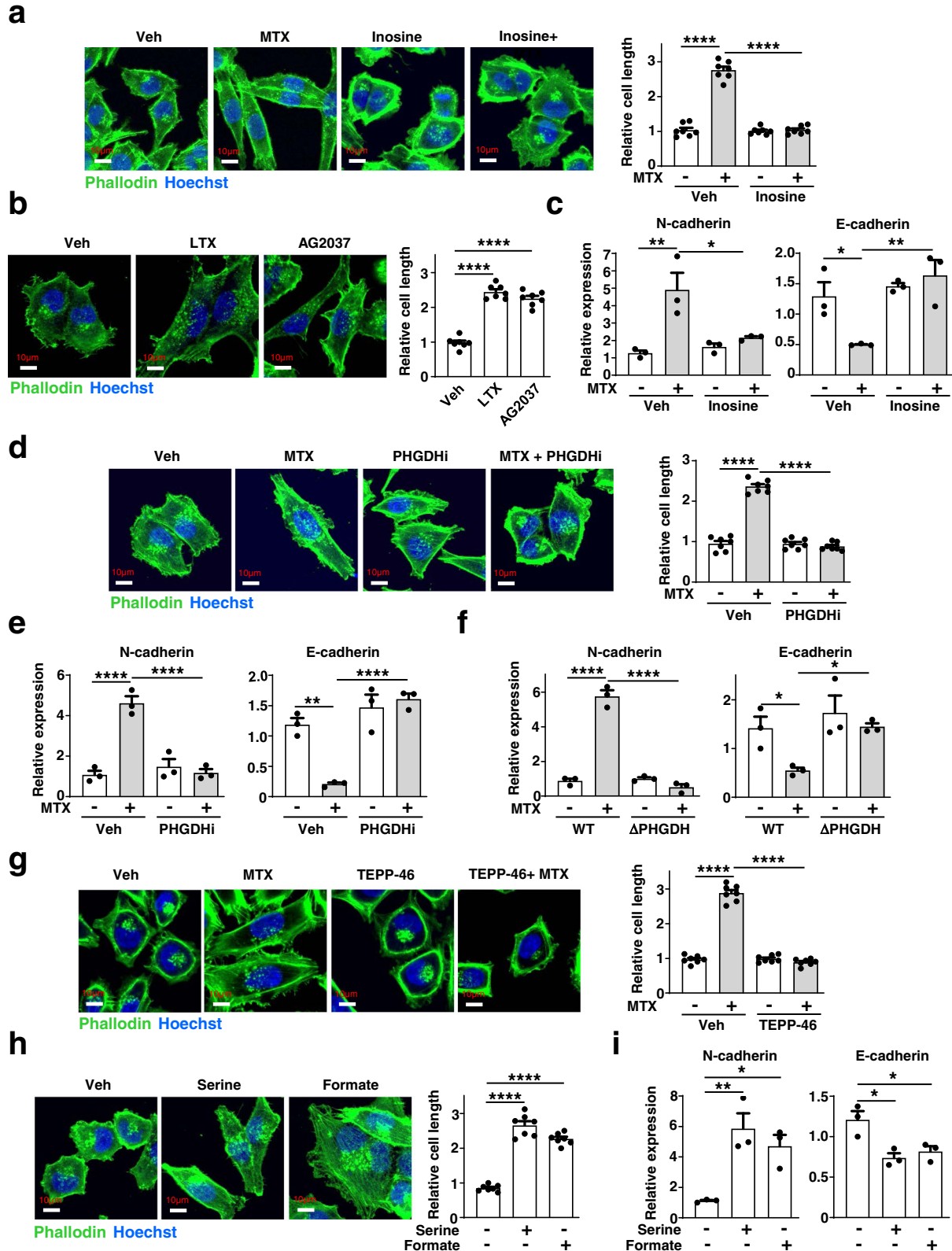

Supermix (Bio-Rad, 1725274) and CFX384 Touch Real-Time PCR Detection System (Bio-Rad). The E-cadherin and N-cadherin data were normalized with the control gene (*RPLP0*). The quantitative PCR data were analyzed with a Bio-Rad CFX Manager version 3.1.1517.0823. Primer sequences for human genes are as below:
Primer sequences:
*ECAD* Forward primer: GCCTCCTGAAAAGAGAGTGGAAG
*ECAD* Reverse: TGGCAGTGTCTCTCCAAATCCG
*NCAD* Forward: CCTCCAGAGTTTACTGCCATGAC

*NCAD* Reverse: GTAGGATCTCCGCCACTGATTC
*VIM* Forward: CTCTCCAAAGGCTGCAGAAGT
*VIM* Reverse: CGCTAAAGCCTGTCTTTGCTC
*RPLP0* Forward: CAGATTGGCTACCCAACTGTT
*RPLP0* Reverse: GGGAAGGTGTAATCCGTCTCC

**Cell lysis, immunoblotting, and antibodies**. Cell lysis was performed with 1% Triton lysis buffer as previously[37]. Cell lysates were clarified by centrifugation

**Fig. 5 Purine depletion triggers cell elongation and an EMT transcriptional program through an increase in cellular serine levels. a** Representative images of actin cytoskeleton staining with Phalloidin (green) and nuclei (blue) from A375 cells treated with MTX (200 nM, 24 h) in the presence or absence of inosine (50 μM, 24 h). Quantification of cell length from seven independent cells is presented relative to vehicle control. **b** As in (**a**), but from A375 cells that were treated with LTX (800 nM, 24 h) or AG2037 (100 nM, 24 h). **c** Relative mRNA expression of the indicated EMT markers upon treatment with MTX (200 nM, 24 h) in the presence or absence of inosine (50 μM, 24 h). **d** As in (**a**), but from A375 cells that were treated with MTX (200 nM, 24 h) in the presence or absence of PHGDH inhibitor (PHGDHi, BI-4924, 15 μM, 24 h). **e** Relative mRNA expression of the indicated EMT markers upon treatment with MTX (200 nM) in the presence or absence of PHGDH inhibitor (PHGDHi, BI-4924, 15 μM, 24 h). **f** Relative mRNA expression of the indicated EMT markers as in **c**, but from wild-type or ΔPHGDH A375 cells treated with either vehicle (DMSO) or MTX (200 nM, 24 h). **g** As in (**a**), but from A375 cells that were treated with MTX (200 nM, 24 h) in the presence or absence of TEPP-46 (100 μM, 24 h). **h** As in (**a**), but from A375 cells that were treated with serine (400 nM, 24 h) or Formate (2 mM, 24 h) in normal DMEM media. **i** Relative mRNA expression of the indicated EMT markers upon treatment with serine (400 nM, 24 h) or Formate (2 mM, 24 h) in normal DMEM media. **a–i** Data are presented as the mean ± s.d from $n = 3$ (**c**, **e**, **f**, **i**) or $n = 7$ (**a**, **b**, **d**, **g**, **h**) of biologically independent samples. **a–i** One-way ANOVA, Turkey's post-hoc test, multiple comparison *$p < 0.05$, **$p < 0.01$, ***$p < 0.001$, ****$p < 0.0001$. **a**, **b**, **d**, **g**, **h**) Scale bars are indicated. Data are representative of $n = 2$ (**c**, **e**, **f**, **i**) or $n = 3$ (**a**, **b**, **d**, **g**, **h**) independent experiments. Source data and exact *p*-values are provided as a Source Data file.

($20{,}000 \times g$ for 15 min at 4 °C), and protein concentrations were measured with Bradford assay (Biorad, 500-0006). Samples (20–30 μg of protein lysate/sample) were subjected to SDS-PAGE followed by immunoblotting using the indicated antibodies primary antibodies: GART (Proteintech, 13659-1-AP, 1:1000), S6 Kinase (CST, 2708, Lot:8, 1:1000), p-S6 Kinase -T389 (CST, 9234, Lot:12, 1:1000), PKM1 (Sigma, SAB4200094, Lot:117M4754V, 1:1000), PKM2 (Cell Signaling, 4053 S, Lot:6, 1:1000), PHGDH (Proteintech, 14719-1-AP, 1:1000), PSAT1 (Proteintech, 10501-1-AP, 1:1000) DHODH (Proteintech, 14877-1-AP, 1:1000), APRT (Abcam, ab196558, 1:1000), HPRT1 (Santa Cruz, sc-393901, 1:1000), MTHFD1L (Proteintech, 16113-1-AP, 1:1000), β-actin (Sigma, A5316, Lot: 059M4770V, 1:5000), and HRP-conjugated anti-mouse (CST, 7076, 1:5000) and anti-rabbit (CST, 7074, 1:5000) secondary antibodies were used.

**Immunohistochemistry.** Lymph nodes and lungs were fixed in 4% paraformaldehyde (PFA) and 10 μm paraffin sections were placed onto Thermo Superfrost slides (Thermo Fisher Scientific, USA). Paraffin processing, embedding, sectioning, and hematoxylin and eosin staining (H&E) were performed by the Histo Pathology Core, UTSW. The H&E section slides were imaged by using an automatic NanoZoomer 2.0-HT (HAMAMATSU, Japan) slide scanner with ×20 mode (0.46 μm/pixel) in our UTSW Whole Brain Microscopy Facility (RRID:SCR_017949). The lung and lymph nodes metastasis from B16 melanoma cancer were quantified by immunofluorescence staining of S100 (S100 antibody, Dako Omnis, Agilent), using a 20× slide scanner Zeiss Axioscan.Z1 (Carl Zeiss, Germany) at the UTSW Whole Brain Microscopy Facility. Melanoma S100 positive cell numbers were quantified by using ImageJ software version Java 1.8.0_172 (National Institutes of Health). A375 lung metastasis was counted from H&E images, using NDP.view 2 software (U12388-0).

**Immunofluorescence microscopy.** Immunofluorescence analysis was performed as previously[38]. Briefly, A375 cells were treated as indicated in the figures and fixed with 4% PFA (Santa Cruz Biotechnology, USA, sc-281692). The actin filaments were stained with Alexa Fluor® 488 Phalloidin (8878 S, Cell signalling technology) for 15 minutes, according to the manufacturers' instructions, and nuclei were stained with Hoechst (Sigma-Merck, Germany, 62249). Images were taken with Zeiss LSM 780 Laser Scanning Microscope (Carl Zeiss, Germany) with 63×/1.4 Plan-Apochromat Objective and cell length was analyzed by using ImageJ software version Java 1.8.0_172 (National Institutes of Health).

**Other reagents.** Methotrexate hydrate (Sigma, A6770-100MG, Lot:BCCD4454), TEPP-46 (MCE, HY#18657, Lot:24615), Lometrexol hydrate (Sigma, SML0040), Adenine hemisulfate salt (Sigma, A2545 Lot:WXBC5060V), BI-4924 (MCE, HY-126254/CS-0101055, Lot:818685), NCT-503 (Selleckchem, S8619, Lot:S861901), PKUMDL-WQ 2101 (Tocris, 6580), Sodium formate (Sigma, 247596), ADP (Sigma, A5285, Lot:SLBZ1414), ATP (Sigma, A2383, Lot# SLBZ3783), DMSO (Sigma, D2650, Lot: RNBJ7906), crystal violet (Sigma, 6158), protease inhibitor cocktail (Sigma, P8340), $^{13}C_6$-glucose (Sigma, 389374), inosine (Sigma, I4125), serine (S4311), glycine (Sigma, G7126), dialyzed FBS (Sigma, F0392), FBS (R&D Systems, S11150), DMEM (Corning/Cellgro, 10-017-CV), glucose-free DMEM (Thermo Fisher Scientific, 11966025), (Polysciences, 24765-1), serine/glycine-free DMEM (Cat #: USBiological, D9802-01), BCA kit (Thermo Fisher Scientific, 23225), and Bradford assay reagent (Bio-Rad) were used as indicated.

**Metabolite profiling.** Metabolite extraction was performed as described previously[39,40]. Briefly, metabolites were extracted in 80% methanol (−80 °C) from nearly confluent cells grown in 10 cm dishes or six-well plates. Cells were scraped in 80% methanol and subjected to centrifugation ($6000 \times g$, 5 min) at 4 °C to isolate the soluble metabolites in the supernatant. The insoluble pellet was subjected to a subsequent extraction with 80% methanol and centrifugation at $21{,}000 \times g$ for

5 min at 4 °C. The supernatants from these extractions were pooled and dried down using an N-EVAP (Organomation Associates, Inc) or in a SpeedVac. Samples were resuspended in 80% acetonitrile prior to running them on an AB QTRAP 5500 (Applied Biosystems SCIEX), as previously[24], or resuspended in water and run on a Q-Exactive mass spectrometer (Thermo Fisher) coupled to a Prominence UPLC system (Shimadzu) with Amide XBridge HILIC chromatography (Waters)[27]. Peak areas from the total ion current for each metabolite SRM transition were integrated using MultiQuant v2.0 software (AB/SCIEX) for samples run on the AB QTRAP 5500. Peak areas for samples run on a Q-Exactive were integrated with TraceFinder 5.1 (ThermoScientific). Heatmaps were presented using MetaboAnalyst5.0 (www.metaboanalyst.ca). Instrumental parameters and software analysis were set as described previously[8,39]. For targeted $^{13}C_6$-Glucose isotopic tracing experiments, cells were seeded in biological triplicate or quadruplicates and incubated with 10mM-$^{13}C_6$-Glucose (CIL, CLM-1396-1) in 10% dialyzed serum in DMEM lacking glucose (Thermo Scientific, 11966025) for the last 1–3 h or as indicated in the figure legends.

**Lung tissue and blood metabolomics.** Ten milligrams of lung tissue were transferred to 2 ml screw-capped Eppendorf tubes containing 1 ml 80% methanol (pre-chilled in −80 °C freezer for at least 20 min) and lysed with Benchmark Bead-Blaster (speed: 3640 m/s, linear speed: 7 m/s, set time: 30 sec, cycles: 3, interrupt time: 10 s, temperature: 4 °C). Tubes were incubated in a −80 °C freezer for 15 min and centrifuged for 5 min at $6000 \times g$ at 4 °C. Supernatants were transferred to a new vial, while the pellet was resuspended with another 500 μl of 80% methanol (second extraction) and centrifuged for 5 min at $20{,}000 \times g$. Both extractions were collected and subjected to another 15 min spin at $20{,}000 \times g$ prior to drying in a SpeedVac. For blood samples, 50 μl of blood was extracted twice with 500 μl of 80% methanol. The supernatants were then dried down in a SpeedVac. The supernatants were resuspended in 80% acetonitrile and run on AB QTRAP 5500 (Applied Biosystems SCIEX). The peak areas were normalized with the protein abundance measured with a BCA assay.

**ADP/ATP concentration measurements by LC/MS.** Cells were cultured in DMEM containing 10% FBS in six-well plates in quadruplicates. Twenty-four hours post-plating, cells were treated with 2 μM MTX for 8 h. Metabolites were extracted in 200 μl of 80% acetonitrile solution (4 °C). Samples were incubated at −80 °C freezer for 15 minutes, transferred into eppendorf tubes, and centrifuged at $15{,}000 \times g$ for 5 min at 4 °C. The supernatants, alongside ADP and ATP standards (range: 10–1600 ng/ml) were run immediately on AB QTRAP 5500 (Applied Biosystems SCIEX)[38]. Intracellular ADP and ATP concentrations were quantified against the standard curve. The number of cells was determined by Trypan Blue. If the cell volume of a single HeLa cell ($Vol_{HeLa1c}$) is ~2500 μm$^3$ ($2.5 \times 10^{-12}$ L)[28], the volume total from which adenylate was extracted can then be calculated as:

$$Vol_{cell}\ total = Vol_{HeLa1c} * Cell\ number$$

The intracellular concentration of adenylate can then be calculated as:

$$[AXP]i = Q_{AXP}/Vol_{cell}total$$

where $[AXP]i$ is the intracellular concentration of ATP or ADP, $Q_{AXP}$ is the quantity in mol of ADP or ATP measured in the reaction, and $Vol_{cell}$ total is the volume total of cells from which adenylates were extracted.

**Extracellular acidification rate (ECAR) and oxygen consumption rate (OCR).** Seahorse XFe96 Analyzer (Seahorse BioScience) was used to assess extracellular acidification rate (ECAR) and oxygen consumption rate (OCR) in cultured cells. HeLa cells were cultured in DMEM supplemented with 10% dialyzed FBS in Seahorse XF96 cell culture microplates (Agilent, V3-PS). The following day, cells were either treated with DMSO, MTX (2 μM), or LTX (2 μM) for 15 h. Prior to OCR measurement, cells were rinsed twice with Seahorse XF media

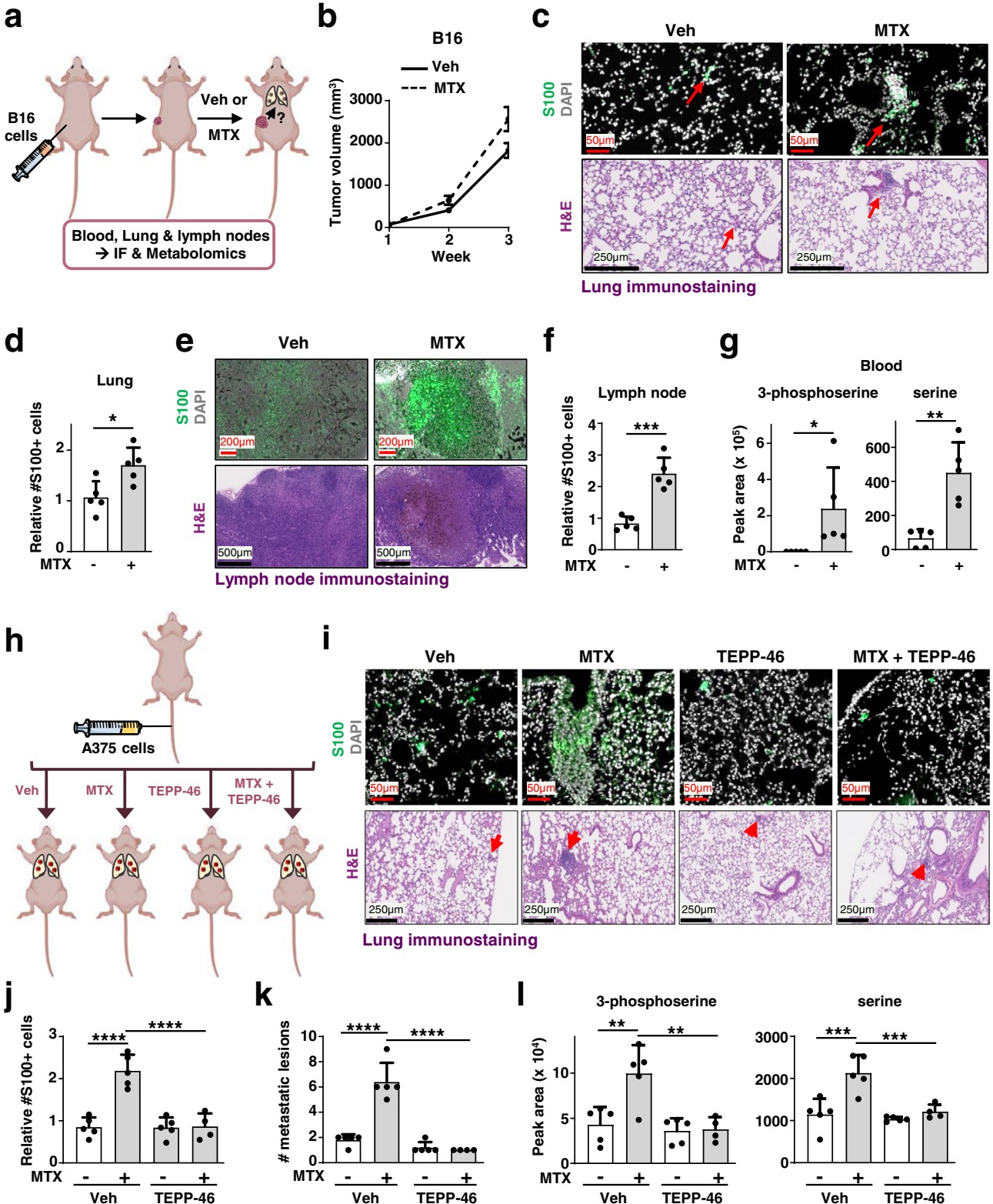

(Agilent # 103575-100, supplemented with 2 mM L-glutamine, 1 mM pyruvate, and 10 mM glucose, pH = 7.4), and incubated for 60 min at 37 °C in a $CO_2$-free incubator. Oligomycin and FCCP were injected for a final concentration of 2 and 1 μM, respectively. Similarly, for ECAR measurement, cells were rinsed with XF media supplemented with 2 mM glutamine but no glucose. The plates were incubated at 37 °C in a $CO_2$-free incubator for 60 min. Oligomycin and FCCP were injected for a final concentration of 2 μM, and 1 μM, respectively. The assay program was set up in Wave software (Agilent Technologies, Version 2.6.1 for Windows), programed for four cycles of 2 min mixing followed by 3 min

measurements. For normalization, cells were stained with Crystal violet solution (Sigma, C6158).

**PHGDH enzymatic assay**. The PHGDH activity assays were adapted from a previous study[41]. The FLAG-tagged PHGDH, PSAT1, and PSPH were purified separately from HEK293E cells[37]. Briefly, HEK293E cells were transfected with plasmids expressing PHGDH, PSAT1, and PSPH using the PEI transfection method. Forty-eight hours post-transfection, cells were lysed in 1% Triton lysis

**Fig. 6 Methotrexate treatment increases the metastatic capacity of melanoma cells in a PKM2-dependent manner. a** Schematic of the metastasis assay depicting subcutaneous injection of B16 melanoma cells. Mice were treated with vehicle (Veh, PBS) or MTX (20 mg/kg, three times a week) once tumors became palpable for three consecutive weeks. **b** Tumor growth of B16 melanoma cells. Data are presented as the mean S.D. from n= 5 animals. **c** Representative images of lung sections immunostained with S100 melanoma marker (green) or hematoxylin-eosin (H&E) from the experiment described in (**a**). **d** Quantification of S100 positive cells relative to vehicle control (PBS) is shown from 5 slides containing lung sections are shown. **e** As in (**c**), but from axillary lymph node stained with S100 melanoma marker (green) or H&E of experiment described in (**a**). **f** Quantification of S100 positive cells relative to vehicle control (PBS) from (**e**) is shown from 5 slides containing lymph sections. **g** Normalized peak areas of 3-phosphoserine and serine from blood. Experiment is described in (**a**). **h** Schematic of a metastasis assay depicting tail-vein injection of A375 cells in athymic nude mice. A375 cells injected into the tail vein of athymic nude mice, treated either with vehicle (PBS), or MTX (20 mg/kg, IP, three times a week), TEPP-46 (50 mg/kg, daily oral gavage), or a combination of MTX with TEPP-46. **i** Representative images of lung sections immunostained with S100 melanoma marker (green) or H&E are shown. $n = 5$ of biologically independent animals, except for $n = 4$ for MTX + TEPP-46 treatment. **j** Quantification of S100 positive cells from lung sections of the experiment in (**h**). **k** Number of metastatic lesions from H&E-stained slides containing lung sections as in (**h**). **l** Normalized peak areas of 3-phosphoserine and serine from lung tissues from experiment described in (**h**). **b**, **d**, **f**, **g**, **j–l** Data are presented as the mean ± s.d from $n = 5$ (**b**, **d**, **f**, **g**, **j–l**) of biologically independent samples. Each data point represents a distinct animal. **j–l** One-way ANOVA, Turkey's post-hoc test, multiple comparison, **b**, **d**, **f**, **g** Unpaired $t$-test, $*p < 0.05$, $**p < 0.01$, $***p < 0.001$, $****p < 0.0001$. **c**, **e**, **i**. Scale bars are indicated. Source data and exact $p$-values are provided as a Source Data file.

buffer. FLAG-tagged proteins were incubated with anti-FLAG M2 Affinity agarose gel for four hours. The agarose beads were washed five times with lysis buffer, once with equilibration buffer (10 mM HEPES, 50 mM NaCl, protease inhibitors) prior to elution with 3xFLAG peptide containing solution (0.2 mg/ml 3xFLAG peptide, 10 mM HEPES, 50 mM NaCl, protease inhibitors). The eluate was passed through 0.45 μm Costar® Spin-X® tube filters, and the purified proteins were subjected to Coomassie staining to determine the purity and the protein amount. 0.5 μg of the purified PHGDH, PSAT1, and PSPH were incubated with the assay buffer (50 mM Tris pH 8.0, 10 mM MgCl2, 0.05% BSA, 0.01% Tween-20, 1.25 mM glutamate, 0.3 mM NAD$^+$ and 3-Phosphoglycerate (0.1 mM)) in a 100 μl reaction in the presence or absence of 1 mM of the indicated nucleotides. Measurements of fluorescence (excitation at 340 nm and emission at 460 nm) were made every 2 minutes at 25 °C with a Perkin-Elmer Enspire Multimode plate reader. The data shown are representative of three independent experiments.

**Cell viability and cell proliferation assays**. A375 (110,000 cells/well) and Hela (80,000 cells/well) cells were plated in triplicates in 12-well plates. Twelve hours post-plating, cells were treated with MTX, PHGDH inhibitors, and PKM2 activator as indicated in the figure legends. Twenty-four or forty-eight hours later, viable cell counts were measured using Trypan Blue with a LUNA-II™ Automated Cell Counter. Counts were normalized to vehicle control (DMSO), and data are presented as relative cell viability.

Cell proliferation was determined with a BrdU Cell Proliferation Assay Kit (CST, 6813) following the manufacturer's instructions. Briefly, A375 cells plated in 96-well plates (10,000 cells/well) were treated with various concentrations of MTX for 24 h and labeled with BrdU for the last 4 h. Absorbance at 450 was measured with SpectraMax iD3 Multimode Plate Detection Platform (Molecular Devices, LLC).

**Wound-healing assay**. The wound-healing (gap closure assay) was adapted from a previous study[42]. Briefly, A375, Cal51, B16, SK-MEL-28, and HeLa cells were seeded in clear-bottom 96-well plates (Corning, 3595) at $3 \times 10^4$ cells/well in 10% DMEM. Confluent cells grown in a monolayer were subjected to overnight (12 h) serum starvation to arrest cell proliferation. For the serine supplementation experiments, cells were serum-starved in serine/glycine-free media (Cat #: USBiological, D9802-01). The scratches were created by a sterile p20 tip, and wells were washed with PBS to remove any detached cells in the opened gap. Two hundred microliters of appropriate media was added in 10% FBS containing DMEM. Supplementation with serine was conducted in serine/glycine-free media. Images were taken at 0 h and 24 h time point with Celigo image cytometer-4 Channel with software 5.1.0 (Nexcelom Bioscience). The gap length was analyzed with ImageJ software version Java 1.8.0_172 (National Institutes of Health), and the data is represented as distance migrated in μM [Gap length at 0 h (minus) gap length at 24 h].

**Transwell (Boyden Chamber) assay**. Hela (400, 000 cells/well) and A375 (600, 000 cells/well) were seeded in six-well plates in DMEM containing 10% FBS. Twenty-four hours post-plating, cells at ~70% confluency were subjected to serum starvation (16 h) with DMEM containing 0.1% BSA in the presence or absence of MTX or other compounds as indicated. Next, cells were trypsinized, counted, and resuspended in DMEM containing 0.1% BSA and treated with agents as indicated in the figure legends. Two hundred microliters of cell suspension containing 50,000–120,000 cells were transferred to the Boyden chamber (upper chamber) (CELLTREAT, 230639), while the wells of the 24-well plates (below the chambers) were filled with 500 μl of 10% FBS. Sixteen hours later, the bottoms of the Boyden chambers were washed with PBS. Using a cotton swab (Q-tips) dipped in PBS, the cells on the top of the Boyden chambers were gently removed, and the migrated cells at the bottom of the chamber were fixed for 15 min in 4% PFA prior to

staining with 0.1% Crystal violet solution (0.1% crystal in 10% ethanol) for 20 min. The Boyden chambers were washed 3× with PBS, and the upper chamber was cleaned with the Q tip again. Boyden chambers were allowed to dry at room temperature for a couple of hours, and the membrane was cut and imaged with Primovert ZEISS microscope with a ×10 objective. All images were recorded with ZEN 3.1 (Blue ed) software and analyzed with ImageJ.

**Preparation of the TP53 guide RNA viral particles**. HEK293T cells were plated at 6 million cells in a 10 cm dish and transfected the following day using 10 μg of lentiviral transfer plasmid, 8 μg of psPAX2 (Addgene #12260), 4 μg of pMD2.G (Addgene #12259), and 44 μg of polyethylenimine (PEI) (Polysciences #23966-1) at a 2:1 ratio of PEI to DNA. Prior to transfection, *TP53* guide DNA and PEI were each separately mixed with 250 μL of 300 mM NaCl, vortexed, combined, and then incubated for 20 min. Transfection complexes were added dropwise to plates and cells were incubated overnight at 37 °C. The following day, the transfection medium was replaced with a fresh medium, and culture supernatant was collected at 48 and 72 h after transfection. The supernatant was centrifuged at $200 \times g$ for 5 min, filtered through a 0.45 μm polyethersulfone (PES) membrane, and then combined with polyethylene glycol (PEG) concentrator solution (40% PEG 8000, 1.2 M NaCl, PBS) at a ratio of 3:1 supernatant to PEG concentrator solution. The supernatant was then incubated overnight at 4 °C to precipitate viral particles and centrifuged at $1500 \times g$ for 45 min. Viral pellets were resuspended in 1 mL of Dulbecco's phosphate-buffered saline (ThermoFisher #14190250) and stored at −80 °C.

**Statistical analysis and software**. Heatmaps were generated with MetaboAnalyst 5.0 (www.metaboanalyst.ca). Statistical analysis was performed using Microsoft Excel 365™ and GraphPad Prism 8.4.3 software. For pairwise comparisons, two-tailed Student's $t$-tests were used, and for multiple comparisons, one-way ANOVA test with Tukey's post-hoc test. All error bars represent standard deviation (S.D.). All experimental data are representative of at least two independent experiments. Two biologically complementary in vivo metastasis experiments were performed. Images of mice for Fig. 2f were obtained from Servier Medical Art (https://smart.servier.com) under a Creative Commons Attribution 3.0 Unported License(https://creativecommons.org/licenses/by/3.0/). The mice in the images in Figs. 6a, 6h were obtained from Biorender (https://biorender.com/). The image in Fig. 4a was assembled in Powerpoint, Microsoft 365®.

**Reporting summary**. Further information on research design is available in the Nature Research Reporting Summary linked to this article.

## Data availability
Source data are provided with this paper. All the data supporting our findings are available in the article, the supplementary files, or the Source data file.

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

## Acknowledgements

We thank Prof. Matthew Vander Heiden (Massachusetts Institute of Technology, Cambridge, MA) for the PKM2 wild-type and knockout MEFs and materials, Marc Morgan and Ali Shilatifard for providing us with the guide RNA targeting human P53, Min Yuan for assistance with mass spectrometry, and Vijayashree Ramesh for assistance with tail-vein injections. This research was supported by grants from the NIH: R01GM143236 (G.H.), R35-CA197459 (B.D.M.), P01-CA120964 (B.D.M. and J.A.), R00-CA194192, R01-GM135587, R01GM143334 (I.B.-S.); a Welch foundation award (I-2067-20210327(G.H.)); a TS Alliance Research Grants Program award (885252 (G.H.)), a LAM Foundation Career Development and Established Investigator Awards (LAM0127C01-18, LAM0151E01-22) (I.B.-S.); a TSC Alliance postdoctoral fellowship (SP0057487) (E.V.); and a Rothberg Courage Award from the TSC Alliance (B.D.M.). A.T. was supported by an Emmy Noether Award from the German Research Foundation (DFG, 467788900) and the Ministry of Culture and Science of the State of North Rhine-Westphalia (NRW-Nachwuchsgruppenprogramm). G.H. is recipient of a CPRIT Scholar (CPRIT; RR190087) and a V Scholar (V2021-019) awards.

## Author contributions

B.D.M., I.B.-S., and G.H. conceptualized, directed the study, and wrote the manuscript. M.H.S., R.K., and U.S. performed and analyzed the data. A.T., E.V., Z.D., D.H.T., E.S.A., H.R., B.P.O., S.K., and J.H.H. assisted with molecular, cell biology, and animal experiments, F.C., H.S.V., M.M., T.P.M., P.G., and J.M.A. performed LC-MS/MS analyses and assisted with mass-spectrometry-related methods. I.B.-S. and G.H. prepared the manuscript.

## Competing interests

B.D.M. is a shareholder and scientific advisory board member of Navitor Pharmaceuticals. The remaining authors declare no competing interests.
