## [Peer Review File · Nature Communications]

Purine nucleotide depletion prompts cell migration by stimulating the serine synthesis pathwayEditorial Note: This manuscript has been previously reviewed at another journal that is not operating a transparent peer review scheme. This document only contains reviewer comments and rebuttal letters for versions considered at *Nature Communications*.

REVIEWERS' COMMENTS

Reviewer #1 (Remarks to the Author):

The authors have more than adequately addressed all of my comments and produced a highly interesting set of findings.

Reviewer #2 (Remarks to the Author):

The authors have done almost all the suggested changes.

The question which is still not fully answered is whether MTX treatment induce invasion or not? The authors provide evidence through literature and speculate about it; but do not show a clear proof in this study. However, with their substantial improvement of the manuscript this issue may be a point of interesting discussion rather than a flaw.

Reviewer #3 (Remarks to the Author):

The authors have addressed my comments and those of the other reviewers in a very thorough way, with new data and changes to the text. I do not have any further requests for changes. I congratulate the authors on a very interesting and timely study that will advance the field and spark new research into connections between metabolism and migration.

We thank the Reviewers for their positive comments. Below, we discuss one lingering comment from Reviewer #2. Reviewers' comments are provided in "*italics*" for reference, with author responses denoted by 'Response:'.

Reviewer #1 (Remarks to the Author):

The authors have more than adequately addressed all of my comments and produced a highly interesting set of findings.

Response: We thank the reviewer for their positive and overall constructive feedback.

Reviewer #2 (Remarks to the Author):

The authors have done almost all the suggested changes.

The question which is still not fully answered is whether MTX treatment induce invasion or not? The authors provide evidence through literature and speculate about it; but do not show a clear proof in this study.

However, with their substantial improvement of the manuscript this issue may be a point of interesting discussion rather than a flaw.

Response: We thank the reviewer for this comment, which we now discuss in the manuscript as below:

Our findings that specific inhibition of purine synthesis with MTX can trigger an EMT transcriptional program, cell migration phenotype in vitro, and enhanced metastatic dissemination in mouse models, suggest that MTX may induce cancer cell invasion. However, further work is required to elucidate whether MTX promotes specific cancer invasion features, enabling tumor cells to overcome barriers of the extracellular matrix and spread into surrounding tissues. Similar to our findings, the support of cancer cell motility, migration, and metastasis needed the increase of de novo serine synthesis and mitochondrial one-carbon metabolism.

Reviewer #3 (Remarks to the Author):

The authors have addressed my comments and those of the other reviewers in a very thorough way, with new data and changes to the text. I do not have any further requests for changes. I congratulate the authors on a very interesting and timely study that will advance the field and spark new research into connections between metabolism and migration.

Response: We thank the reviewer for their positive comments on our manuscript.